# Solvent Etching Process for Graphitic Carbon Nitride Photocatalysts Containing Platinum Cocatalyst: Effects of Water Hydrolysis on Photocatalytic Properties and Hydrogen Evolution Behaviors

**DOI:** 10.3390/nano12071188

**Published:** 2022-04-02

**Authors:** Thi Van Anh Hoang, Thi Kim Anh Nguyen, Duc Quang Dao, Phuong Anh Nguyen, Dong Hwi Jeong, Eun Woo Shin

**Affiliations:** School of Chemical Engineering, University of Ulsan, Daehakro 93, Nam-gu, Ulsan 44610, Korea; hoangvananh2203@gmail.com (T.V.A.H.); nguyenthikimanhtb@gmail.com (T.K.A.N.); ddquang221@gmail.com (D.Q.D.); anhphuong.nguyen1150@gmail.com (P.A.N.); jdonghwi@ulsan.ac.kr (D.H.J.)

**Keywords:** solvent etching, Pt/g-C_3_N_4_, tri-s-triazine unit, water content, hydrolysis, photocatalytic hydrogen evolution

## Abstract

In this study, we synthesized Pt/g-C_3_N_4_ photocatalysts modified by a solvent etching process where ethanol (Pt/CN0), water (Pt/CN100), and a 50:50 mixture (Pt/CN50) were used as a solvent, and investigated the optimal properties of g-C_3_N_4_ to prepare the best Pt/g-C_3_N_4_ for photocatalytic hydrogen evolution. From diverse characterizations, water was proven to be a stronger solvent agent, resulting in not only the introduction of more O-functional groups onto the g-C_3_N_4_ surface, but also the degradation of a regular array of tri-s-triazine units in the g-C_3_N_4_ structure. While the addition of O-functional groups positively influenced the oxidation state of the Pt cocatalyst and the hydrogen production rate, the changes to g-C_3_N_4_ structure retarded charge transfer on its surface, inducing negative effects such as fast recombination and less oxidized Pt species. Pt/CN50 that was synthesized with the 50:50 solvent mixture exhibited the highest hydrogen production rate of 590.9 µmol g^−1^h^−1^, while the hydrogen production rates of Pt/CN0 (with pure ethanol solvent) and Pt/CN100 (with pure water solvent) were 462.7, and 367.3 µmol g^−1^h^−1^, respectively.

## 1. Introduction

With the rising need for sustainable industrial systems, the replacement of fossil fuels with renewable energy resources has attracted considerable attention [1]. As a renewable energy resource, hydrogen from photocatalytic water-splitting based on semiconductors has the potential to meet the green energy mandate of little or no CO_2_ emissions [2,3,4]. To date, a variety of semiconductor materials have been employed to simultaneously drive water oxidation and reduction [5,6,7,8,9]. Among a vast number of semiconductor photocatalysts, graphitic carbon nitride (g-C_3_N_4_), a metal-free organic polymerized material, has been widely utilized in interdisciplinary hydrogen evolution, owing to its excellent photocatalytic characteristics, including low-cost precursors, chemical stability in the ambient environment, sufficient band gap for solar light usage, and ecological friendliness [10,11]. Nevertheless, bulk g-C_3_N_4_ prepared through direct polycondensation of nitrogen-rich precursors presents several obstacles that limit its practical applications: low specific surface area, fast recombination of photogenerated charge carriers, and its hydrophobic surface [12,13,14]. In the past decade, many researchers have made immense efforts to overcome these drawbacks, including heteroatom doping [15], heterostructure construction [16,17,18], defects engineering [19,20,21,22,23], and vacancy formation [24].

Recently, a solvent etching process has been introduced as an effective, uncomplicated method to significantly improve the photocatalytic performance of g-C_3_N_4_ materials [25,26,27,28,29,30,31]. In the solvent etching process, water is a common solvent that has a great impact on increasing the specific surface area and attaching enormous O-containing functional groups onto the g-C_3_N_4_ surface [26,27,28]. Moreover, nitrogen deficiency in g-C_3_N_4_ induced by the water solvent etching plays an important role in efficient electron-hole separation, resulting in an improvement of the photocatalytic activity [27,31]. However, the strong hydrolytic property of water incidentally changes the morphological structure of g-C_3_N_4_ and leads to the intense deformation of tri-s-triazine units, which prevents the best H_2_ evolution performance [29,31,32]. Consequently, it is necessary to weaken the hydrolytic property of the solvent to efficiently introduce O-containing functional groups onto the g-C_3_N_4_ surface without the morphological degradation of the g-C_3_N_4_ structure. The addition of an oxidant into the water solvent etching process enhances the degradation of tri-s-triazine units as well as the exfoliation of the g-C_3_N_4_ structure [27,33,34].

Without any oxidant, water-only solvent etching has been conducted through hydrothermal methods [31,32]. In research by Ming et al. [32], even though pure water was employed for hydrothermal solvent etching, morphological degradation of the g-C_3_N_4_ structure was still observed, indicating that the hydrolytic property of water was still strong in the hydrothermal condition. Nguyen et al. [30] suggested ethanol as a new weak solvent to mildly modify the g-C_3_N_4_ surface properties in the solvent etching process. Through that ethanol solvothermal treatment, the surface properties of the g-C_3_N_4_ were weakly modified, with an increase in the photocatalytic evolution rate by a maximum of 25.6%. Based on these observations and our knowledge, we believe that mixing the two solvents—ethanol and water—can control the hydrolysis in the solvent etching process and efficiently enhance the photocatalytic performance of g-C_3_N_4_ materials.

In this study, we modified bulk g-C_3_N_4_ material via the solvent etching process with pure water, pure ethanol, and a mixture of water and ethanol 50:50 (by volume) as the solvent. Platinum (Pt) was impregnated as a cocatalyst into the modified g-C_3_N_4_ material through the photodeposition method. The prepared Pt/g-C_3_N_4_ photocatalysts were characterized by various instruments to measure their physicochemical, morphological, and optical properties. Eventually, photocatalytic H_2_ evolution tests were conducted under visible-light irradiation to confirm each solvent’s hydrolytic effect on the photocatalytic H_2_ evolution rate. In comparison with pure water and pure ethanol, the mixture solvent efficiently modified the Pt/g-C_3_H_4_ photocatalyst so that it exhibited excellent photocatalytic H_2_ evolution. We demonstrated that the moderate mixed-solvent hydrolysis facilitated the creation of the largest specific surface area, the formation of desired O-functional groups on the g-C_3_N_4_ surface, and the highest Pt^2+^ proportion, whereas the strong hydrolytic effect of pure water degraded the regular array of the tri-s-triazine units of the g-C_3_N_4_, resulting in a lower separation efficiency of the photoexcited charge carriers due to the morphological change in the g-C_3_N_4_.

## 2. Materials and Methods

### 2.1. Synthesis of CNx

Thiourea (CH_4_N_2_S, ≥99%), triethanolamine (TEOA, C_6_H_15_NO_3_, 99%), and chloroplatinic acid hexahydrate (H_2_PtCl_6_·6H_2_O) were supplied by Sigma-Aldrich Korea (Gyeonggi, South Korea). Ethanol (C_2_H_5_OH, 99.9%) was acquired from Daejung Chemicals & Metals Co., Ltd. (Gyeonggi, South Korea). All chemicals were of analytical grade and were used without any further purification. Deionized (DI) water was used as a solvent in hydrothermal treatments, sample washing, and photocatalytic reactions. Bulk g-C_3_N_4_ was synthesized by a thermal polymerization method [35]. Typically, 6 g of thiourea precursor was placed in a B-form crucible covered with aluminum foil and heated to 550 °C for 4 h (ramping rate = 5 °C/min) in a muffle furnace. The obtained solid was cooled to room temperature and ground for 15 min in a mortar. The prepared bulk g-C_3_N_4_ sample was designated as CN.

One gram of bulk CN powder was put into 100 mL of pure water, pure ethanol, or a DI/ethanol 50:50 mixture, followed by sonification for 2 h at ambient temperature. The CNx samples (x = 0, 50, or 100 where x was the content of water in the solution mixture) were transferred into a 120-mL Teflon™ autoclave, which was heated at 180 °C for 6 h (ramping rate = 5 °C/min) in an oven. After cooling to room temperature, the obtained solid was filtered and washed several times with DI water.

### 2.2. Synthesis of Pt/CNx

The CNx samples (0.5 mg/L) were first dispersed in 100 mL of DI water by a magnetic stirrer for 30 min. Then a calculated amount of H_2_PtC_l6_.6H_2_O was added so as to establish the initial Pt content as 3 wt%. After purging with pure argon (Ar) for 20 min, the photodeposition of Pt on the CNx was carried out in 60 min under solar-simulated irradiation. In the last step, the precipitates were collected, washed with DI water several times, and freeze-dried for further characterization. The procedure details are described elsewhere [25,28]. The obtained materials were labeled Pt/CN0, Pt/CN50, and Pt/CN100, corresponding to their CNx precursors.

### 2.3. Characterization

An inductively coupled plasma-optical emission spectrometer (ICP-OES) (700-ES; Varian Australia Pty. Ltd., Mulgrave, Australia) was used to determine the Pt element composition of the Pt/CNx samples. The Brunauer–Emmett–Teller (BET) specific surface areas were measured by nitrogen adsorption-desorption isotherms and a QUADRASORB^TM^ SI surface area instrument (Quantachrome Instruments, Boynton Beach, FL, USA). Scanning electron microscopy (SEM) (JSM-600F; JEOL Ltd., Tokyo, Japan) and transmission electron microscopy (TEM) (JEM-2100F; JEOL Ltd., Tokyo, Japan) were applied to study the morphologies of the Pt/CNx and the distribution of Pt nanoparticles. The X-ray diffraction (XRD) (D/MAZX 2500 V/PC high-power diffractometer; Rigaku, Tokyo, Japan) patterns of as-prepared samples were analyzed at a scan rate of 2° (2θ)/min with a Cu Kα X-ray source and a wavelength of λ = 1.5415 Å. The functional groups of the prepared photocatalysts were identified using a Fourier transform infrared (FTIR) transmittance spectrometer (Nicolet™ 380 spectrometer, Nicolet™ iS5 with an iD1 transmission accessory; Thermo Scientific™, Waltham, MA, USA). X-ray photoelectron spectroscopy (XPS) was conducted with a Thermo Scientific’s K-Alpha system (Waltham, MA, USA). Optical properties of the photocatalysts were investigated using a UV-Vis diffuse reflectance spectrometer (SPECORD^®^ 210 Plus spectroscope; Analytik Jena, Jena, Germany) and photoluminescence (PL) measurements (Cary Eclipse fluorescence spectrophotometer; Agilent, Santa Clara, CA, USA). PL spectra were monitored at ambient temperature with a 473-nm diode laser. In addition, the electrochemical impedance spectroscopy (EIS) data was measured on an impedance analyzer (VSP series; Bio-Logic Science Instruments, Seyssinet–Pariset, France). A frequency range of 0.01 Hz to 100 kHz with a 10-mV amplitude and a direct current potential of +0.8 VSCE was used after a 10-min delay under a 3-W visible light bulb illumination. Twenty milligrams of mortar-ground catalyst powder was mixed with 2 mg of active carbon for 20 min to obtain a fine powder. Then the powder was added into 100 µL isopropanol 99.7% and 30 μL nafion 5 wt% (both from Sigma-Aldrich Korea, Gyeonggi, Korea). The electrolyte in the three-electrode setup was 10 mL of sodium hydroxide 1 M (NaOH) solution. A RE-1BP (Ag/AgCl) electrode was the reference electrode and the counter-electrode was a platinum wire. The working electrode was a 6-mm standard-type glassy carbon electrode upon which 10 μL of the sample (1 μL of suspension each time by micro-pipette) was placed.

### 2.4. H_2_ Evolution Test

The photocatalytic H_2_ evolution tests were conducted in a solar simulation system utilizing an LED lamp as the light source, and the light intensity on the solution was 100 mW cm^−2^. In total, 50 mg of each photocatalyst was dispersed in 90 mL of DI water for 30 min with high-purity Ar purging in darkness. Then, 10 mL of TEOA was added to the solution as a hole scavenger. The reactor system was purged with Ar before the H_2_ evolution test began. The gas products were analyzed by an on-line gas chromatograph equipped with a thermal conductivity detector (TCD). The photocatalyst stability tests were carried out under the same setting conditions in 5-h cycles. An excessive amount of TEOA (10 mL) was added to the solution at the start of each cycle, and the reactor system was purged again with Ar for 20 min.

## 3. Results and Discussion

### 3.1. Physicochemical and Structural Properties

In Table 1, the specific surface areas (S_BET_), pore volumes (V), and average pore diameters (L) of the prepared Pt/CNx samples are presented. Appendix A shows the corresponding N_2_ adsorption-desorption isotherms of all the prepared samples. The pattern of the isotherms can be classified as IUPAC type IV containing the H3 hysteresis loop, proving the mesoporous structure of the materials. Interestingly, in Table 1, with the 50:50 mixture of ethanol and water solvent, Pt/CN50 reveals the largest S_BET_ (70.5 m^2^/g) even though this treatment increases the S_BET_ values of the g-C_3_N_4_, indicating that the treatment affects the textural properties. Also, the average pore size for the Pt/CN100 suddenly increases up to 24 nm, being similar to that of the Pt/CN (27.8 nm), implying that the pure water treatment can change not only surface area but also the morphology of the bulk g-C_3_N_4_. This phenomenon can be related to the effect of each solvent on the g-C_3_N_4_ surface structure during the solvent etching, which was also verified via the SEM/TEM images.

The morphological structures of the Pt/CN, Pt/CN0, Pt/CN50, and Pt/CN100 materials were examined with the SEM and TEM images illustrated in Figure 1. Being dissimilar to the bulk g-C_3_N_4_ that was made up of largely agglomerated blocks (Figure 1A,E), the Pt/CN0 for which pure ethanol was used as the solvent exhibits a flake-like structure (Figure 1B) with clean, thin layers (Figure 1F), demonstrating the weak solvent etching ability. In contrast, due to a devastating characteristic of water as a strong solvent that severely attacks the g-C_3_N_4_ structure, the Pt/CN100 contains much smaller nanoflakes with undulating folds at the edges (Figure 1D) where a silk-like surface with clean, porous defects is displayed (Figure 1H). Moreover, from the TEM image (Figure 1G), it is undeniable that the combined ethanol/water solvent in the etching process generated an exceptional morphological characteristic—the irregular nanoporous defects on the surface of Pt/CN50—regardless of it having the same flake-like structure as the Pt/CN0 (Figure 1B,C).

The Pt/CN50 inherits the balance between delaminating to form thinner layers and generating enormous porous defects on the surfaces, resulting in the largest specific surface area with irregular porous defects. The distributions of the C, N, O, and Pt components on the surface of the Pt/CN50 were observed via EDS elemental mappings (Figure 1I), which clearly proved the existence of Pt on the surface of the Pt/CN50. Even though the TEM images of the Pt/CN and Pt/CN0 could not provide the Pt particle size distribution due to no detection or limited detection of the highly dispersed Pt nanoparticles, the TEM images of Pt/CN50 and Pt/CN100 clearly show the Pt nanoparticles and their size distributions (Appendix A). Most of the Pt particle sizes are 2–3 nm on the Pt/CN50 and 3–4 nm on the Pt/CN100, indicating no great difference.

The multilayer graphitic structure of the prepared Pt/CNx catalysts was evidenced by the XRD patterns (Figure 2A). All of the photocatalysts have two interference patterns, and each peak in the patterns is quite similar to those of the pristine g-C_3_N_4_. The typical characteristic XRD peaks appearing at roughly 2θ = 13° and 27.6° can be assigned as the (100) and (002) crystal planes of g-C_3_N_4_, respectively (JCPDS 87-1526) [36]. The tenuous diffraction peak at 13° is ascribed to the in-layer packing structure of a tri-s-triazine unit (melem) in the g-C_3_N_4_, while the intense peak at 27.6° corresponds to the interplanar staking in the g-C_3_N_4_ [33,37,38,39]. These distinguishing peaks at 2θ = 13° and 27.6° indicate the remaining g-C_3_N_4_ crystalline structure of the Pt/CNx even after the solvent etching. However, it is interesting and important that a new peak at 10.6° assigned to a melem-containing structure appears, and the corresponding in-planar peak at 13° almost disappears for the Pt/CN100 [32,40]. From the calculation using Bragg’s equation, the in-planar distance of the melem-like structure increases from 0.68 nm for the other photocatalysts to 0.83 nm for the Pt/CN100 [4].

Figure 3 illustrates the morphological changes in the Pt/CNx photocatalysts by solvent etching. The strong hydrolysis/oxidation effect of water on the g-C_3_N_4_ structure may cause severe changes in the morphology of the g-C_3_N_4_ surface [32]. As a result, the g-C_3_N_4_ structure is strongly degraded and deformed to a melem-like structure deformation in the Pt/CN100 during the solvent etching. The melem units partially detach from the g-C_3_N_4_ structure and more hydroxyl and carboxyl groups are generated on the surface, which results in the expansion in the unit length and the development of clean, porous defects on the g-C_3_N_4_ surface [26]. Although the ethanol/water solvent for the Pt/CN50 was not strong enough to cause these changes during the solvent etching process, there was a slight change in the morphological g-C_3_N_4_ structure—tiny, irregular, porous defects were created.

In the FTIR spectra for all the photocatalysts (Figure 2B), besides an intense band at 808 cm^−1^ representing a breathing mode of the heptazine unit, the absorbance bands in the range of 1200–1600 cm^−1^ were monitored and assigned to the typical stretching vibration modes of the aromatic structure of C–N heterocycles [41,42]. The most interesting bands in Figure 2B are the broad bands appearing at 3000–3600 cm^−1^, corresponding to the stretching modes of N-H bonding or the hydroxyl groups [33,43]. In comparison with the FTIR spectrum of the Pt/CN catalyst, the FTIR spectra of the other Pt/CNx photocatalysts show broader bands having broad shoulders around 3200–3600 cm^−1^, which provides evidence that OH functional groups are additionally generated on the g-C_3_N_4_ surfaces during the solvent etching process. To quantitatively compare the formation of those OH functional groups, the height ratios of the band at 3300 cm^−1^ (representing a typical N–H band) to the band at 3400 cm^−1^ (representing a typical O–H band) were calculated. The ratios were 4.4, 1.3, 1.3, and 1.2 for the Pt/CN, Pt/CN0, Pt/CN50, and Pt/CN100, respectively. The decrease in the height ratios apparently indicates enrichment of the –OH groups in the g-C_3_N_4_ structure due to the strong ability of water to introduce the functional groups during the solvent etching process.

The chemical bonding states of all the photocatalysts were characterized by XPS measurements, and the XPS data of C 1s, N 1s, O 1s, and Pt 4f for all the photocatalysts are plotted in Figure 4. First, the C 1s XPS data of the prepared photocatalysts were deconvoluted into four peaks at 289.0 eV, 287.9 eV, 286.6 eV, and 284.4 eV. The peaks at 287.9 eV and 286.6 eV correspond to the sp^2^-bonded carbon (N–C=N) and the carbon species with –NH_x_ groups, respectively, while a binding energy peak at 284.4 eV represents the standard graphitic carbon C(sp^2^)–C(sp^2^) [33,37,44]. In addition, the formation of a peak at 289.0 eV for the Pt/CNx photocatalysts is related to the –COOH species generated during the solvent etching process [33]. The XPS data of N 1s for all the photocatalysts were fitted to three prominent deconvoluted peaks centered at 400.1 eV, 399.0 eV, and 397.9 eV, which were assigned as the binding energies of the amino groups, the tertiary nitrogen of N–(C)_3_, and the C–N=C in the aromatic system [37,45]. In contrast, O 1s XPS data for Pt/CN displays a broad peak centered at 531.5 eV that is assigned to the C=O species. The solvent etching process introduced two more oxygen functional groups on the g-C_3_N_4_ surface, and the O 1s spectra for the Pt/CNx photocatalysts show additional new peaks centered at 533.0 eV and 530.4 eV corresponding to the appearance of –OH and –COOH groups after the solvent etching, respectively [33]. The addition of O-containing functional groups in all the Pt/CNx photocatalysts can be confirmed by the EA results. Table 2 presents the weight percentages (wt%) of C, N, and O obtained from the ICP-OES measurements. The ratios of O to N for Pt/CN0, Pt/CN50, and Pt/CN100 are 0.008, 0.029, and 0.068, respectively, reflecting an increase in the O wt% along with the water content in the solvent for the solvent etching.

The distributions of the O-containing functional groups on the g-C_3_N_4_ surface were calculated from the XPS O 1s data. The atomic percentages (at%) of C=O, C–OH, and COOH for the Pt/CNx photocatalysts are presented in Figure 5A and Appendix A. The at% values of C=O for Pt/CN0, Pt/CN50, and Pt/CN100 are 37.0%, 26.9%, and 18.2%, respectively. As mentioned before, the single C=O species on the g-C_3_N_4_ surface of Pt/CN is partially converted into C–OH and COOH during the solvent etching process, gradually decreasing the at% of C=O along with the water content in the solvent, which confirms again that water is a stronger solvent etching agent than ethanol. Consequently, with the additional oxygen contents and the higher at% of the –COOH and –OH groups, the Pt/CN100 contained the highest surface concentration of the –COOH and –OH groups on the g-C_3_N_4_ surface. To quantitatively compare the proportions of these O-functional groups in the Pt/CNx photocatalysts, Pt/CN100 shows a higher combined at% of the –COOH and –OH groups on the g-C_3_N_4_ surface (81.8%) than Pt/CN50 (73.1%), and Pt/CN0 (63.0%) in Figure 5A. In our previous study [27], the –COOH and –OH groups were clearly identified for photoexcited electron storage that improved the charge separation rate. According to those results, the negatively charged Pt precursors strongly interacted with positively charged locales on the g-C_3_N_4_ surface containing the O-functional groups and were mainly reduced to PtO during the photodeposition. Based on those observations, it is predicted that Pt/CN100 should contain the highest Pt^2+^ concentration. In fact, the trend in the combined at% of the –COOH and C–OH groups in this study was consistent with the results from the FTIR spectra that showed the increase in –OH groups along with the water content in the solvent. Moreover, the amount of the –COOH and –OH groups in the Pt/CN100 was greater than those in the Pt/CN0 and Pt/CN50, since the O content in the Pt/CN100 (4.71 wt%) was approximately eight times greater than that in the Pt/CN0 (0.61 wt%), as shown in the EA data (Table 2).

The chemical states of Pt in all the photocatalysts were investigated via the XPS data of Pt 4f, as shown in Figure 4. From the deconvolution, three pairs of doublets for Pt 4f were monitored for all the photocatalysts: doublet peaks that appeared at 78.2 eV and 74.9 eV were assigned to Pt^4+^ 4f_5/2_ and Pt^4+^ 4f_7/2_, respectively, which indicated an incomplete reduction of Pt species; doublet peaks at 76.1 and 72.7 eV were assigned as Pt^2+^ 4f_5/2_ and Pt^2+^ 4f_7/2_, respectively, which represented PtO; and doublet peaks at 74.5 and 71.2 eV corresponded to the Pt^0^ metallic phase [27,46]. The atomic percentage ratios of the Pt species for the Pt/CNx catalysts are shown in Figure 5B and Appendix A. Interestingly, Pt/CN50 contains the highest Pt^2+^/Pt (0.57) and the lowest Pt^4+^/Pt (0.25) ratios compared to Pt/CN0 and Pt/CN100, which was different from the expectation based on the distribution of the –COOH and C–OH groups on the g-C_3_N_4_ surface. In this study, the Pt precursor was reduced via the photodeposition method, where the transfer rate of the photoexcited electrons is essential in the reduction step of the Pt precursor. The charge transfer rates on the g-C_3_N_4_ surface were influenced by not only the distribution of the functional groups on the g-C_3_N_4_ surface, but also the morphological structure of the g-C_3_N_4_. For example, the degradation of the regular arrays of the melem units in the Pt/CN100 could retard the charge transfer on the g-C_3_N_4_ surface. Accordingly, the optical properties of the photocatalysts had to be measured to understand the relationship between the optical properties and the photocatalytic behavior.

### 3.2. Optical Properties

The UV-Vis diffuse reflectance spectra were monitored to investigate the optical characteristics of the as-prepared samples (Figure 6A). The band gap values for the Pt/CNx photocatalysts were calculated using Tauc plot spectra produced from the UV-Vis spectra (Figure 6B) and are listed in Table 1. All the Pt/CNx samples revealed prominent absorption peaks at 320 nm (the UV region) and a broad absorption tail into the visible-light region. From the Tauc plot, the band gap values for Pt/CN50 and Pt/CN100 were the same at 3.11 eV, and Pt/CN0 had a band gap absorption value of 2.72 eV. The photon absorbance and band gaps of Pt/CN50 and Pt/CN100 were higher than those of Pt/CN0, and could be related to the strong exfoliation effect of water and the supplement of the –OH functional groups in the presence of the water solvent [47]. Nonetheless, the band gap absorption results of the Pt/CN50 and Pt/CN100 were valid enough to maintain the photocatalytic activity under the visible-light irradiation [48].

The PL emission spectra of the Pt/CNx catalysts were obtained to characterize the photogenerated electron-hole pair separation efficiency. As shown in Figure 6C, all of the PL spectra for the Pt/CNx catalysts exhibit the same emission patterns, showing two main emission bands centered at 435 nm and 460 nm, which is in good agreement with our previous results [30]. Since the emission intensity of PL spectra is proportional to a recombination rate of photoexcited electron-hole pairs, it is considered as an indirect index to estimate the recombination rate of photocatalysts [49]. In Figure 6C, the emission intensities in the PL spectra follow the order of Pt/CN100 > Pt/CN50 ≈ Pt/CN0. That is, Pt/CN100 has the highest emission intensity, resulting in the fastest rapid recombination rate. In comparison, the PL emission intensities for Pt/CN0 and Pt/CN50 are similar. However, the electron-hole pair recombination rate for Pt/CN50 is slower than that for Pt/CN0 because, in the cycle of photoexcitation and relaxation, the absorption intensity for Pt/CN0 is higher than that for Pt/CN50 even though the two photocatalysts have very similar emission intensities.

To evaluate the charge separation efficiency of the Pt/CNx photocatalysts, EIS Nyquist plots were employed, as illustrated in Figure 6D. A smaller semicircle arc radius in the EIS spectra reflects a lower charge transfer resistance [50]. In Figure 6D, the smallest circle radius belongs to Pt/CN50, highlighting that the combination of water and ethanol significantly intensified the electronic conductivity on the g-C_3_N_4_ surface—minimizing the charge transfer resistance. The diameter of the semicircle arc for Pt/CN0 ranges around 25 kΩ of the Z value, which is similar to the values reported in our previous study [30]. Furthermore, and surprisingly, the largest semicircle arc diameter of 40 kΩ for Pt/CN100 demonstrates that the charge separation efficiency over Pt/CN100 significantly decreased despite it having the most O-containing functional groups. As seen in Figure 3, the extension and degradation of an array of the melem units in Pt/CN100 induced an increase in the electrical resistance on the g-C_3_N_4_ surface, resulting in the fast recombination of the photogenerated electron-hole pairs and the slower conduction of the photoexcited electrons. Consequently, this characteristic of the charge separation efficiency of Pt/CN100 may result not only in a decrease in its photocatalytic activity in H_2_ evolution but might also influence the oxidation states of the Pt species during the photodeposition process.

### 3.3. Hydrogen Evolution Tests

Figure 7A shows the photocatalytic H_2_ evolution over time for all the catalysts based on the water-splitting process. The photocatalytic activity tests were conducted under visible-light irradiation (λ > 400 nm). The Pt/CN0 and Pt/CN100 photocatalysts that employed 100% ethanol and 100% water as the solvent agent, respectively, manifested minor H_2_ production of 2313 μmol g^−1^ and 1837 μmol g^−1^ after 5 h of irradiation (Figure 7A). However, the 50:50 combination of ethanol and water promoted the photocatalytic activity at the highest H_2_ production of 2954 μmol g^−1^. The H_2_ evolution rates of Pt/CN0, Pt/CN50, and Pt/CN100 were greater than that of Pt/CN (330.2 μmol g^−1^h ^−1^). Figure 7B quantitatively displays the H_2_ evolution rates of all of the photocatalysts, which increased in the order of Pt/CN100 < Pt/CN0 < Pt/CN50 (367.3, 462.7, and 590.9 μmol g^−1^h ^−1^, respectively).

The volcano-shaped trend in the photocatalytic H_2_ evolution rates can be explained by the synergistic effect of the water/ethanol mixture as the solvent agent on the structural properties of the g-C_3_N_4_ surface and the oxidation state distribution of the cocatalyst Pt. Even though a mixture of ethanol and water is the best solvent to modify the g-C_3_N_4_ surface in the solvent etching process, the origin of the improvement in the photocatalytic performance can be described in terms of the hydrolytic property of water. Figure 8 illustrates the key properties of the Pt/CN50 and Pt/CN100 in each preparation step. The solvent etching process modifies the regular melem unit of g-C_3_N_4_ by the introduction of the O-functional groups, and the degradation of the melem unit via hydrolysis by water. For Pt/CN50, the moderate hydrolysis in the solvent etching process changes the distribution of the O-functional groups on the g-C_3_N_4_ surface and generates tiny, irregular porous defects with no extension of the melem unit. However, in the case of Pt/CN100, the strong water hydrolysis destroys the regular array of the melem units on the g-C_3_N_4_ by extending the melem unit from 0.68 nm to 0.83 nm and generating large, clean holes on the g-C_3_N_4_ surface. The degraded melem unit array in the Pt/CN100 might retard the charge transfer on the g-C_3_N_4_ surface, resulting in not only a decrease in the photocatalytic activity but also a less-oxidized Pt state. The Pt/CN50 has higher proportions of PtO (57.4%) and metallic Pt (17.3%) than the Pt/CN100 (53.9% for PtO and 14.3% for Pt). The Pt^4+^ species is inactive in the photocatalysis, and PtO is considered to be better than metallic Pt as an active site for H_2_ evolution due to its favorable H_2_O adsorption and irreversible process [27,51]; the higher proportion of PtO in Pt/CN50 causes its better photocatalytic H_2_ evolution performance. However, the CN100 photocatalyst containing no Pt species was employed for the photocatalytic hydrogen evolution test in comparison and it had no photocatalytic activity due to the absence of Pt species (Appendix A).

In summary, the moderate hydrolysis by the 50:50 ethanol/water solvent mixture over the Pt/CN50 during the etching process resulted in an efficient photoinduced charge transfer and a high content of Pt^2+^ on the g-C_3_N_4_ surface, which promoted its photocatalytic H_2_ production activity. The reusability and stability of Pt/CN50 were also investigated due to their crucial roles in practical applications. As shown in Figure 7C, after four photocatalytic cycles, the Pt/CN50 displayed no conspicuous decrease in the H_2_ production rate, reflecting its high stability during photocatalytic activity.

## 4. Conclusions

In this study, a facile and eco-friendly 50:50 ethanol/water solvent efficiently modified g-C_3_N_4_ surface properties in a solvothermal process and successfully induced an optimal oxidation state of the Pt cocatalyst, resulting in the best photocatalytic H_2_ evolution performance. The moderate hydrolysis effect of the solvent mixture generated ample O-containing functional groups on the pristine g-C_3_N_4_ surface with no destruction in the regular array of the melem units, which corresponded to efficient charge transfer on the g-C_3_N_4_ surface. This characteristic contributed to the formation of the Pt^2+^ oxidation state in the photodeposition process and eventually enhanced the H_2_ evolution rate by the slow recombination of electron-hole pairs, favorable H_2_O adsorption, and an irreversible reduction process on PtO. Consequently, Pt/CN50 had the highest photocatalytic activity for H_2_ evolution, at 590.9 µmol g^−1^h^−1^, due to the synergistic effect of the moderately hydrolytic ethanol/water solvent.

## Figures and Tables

**Figure 1 nanomaterials-12-01188-f001:**
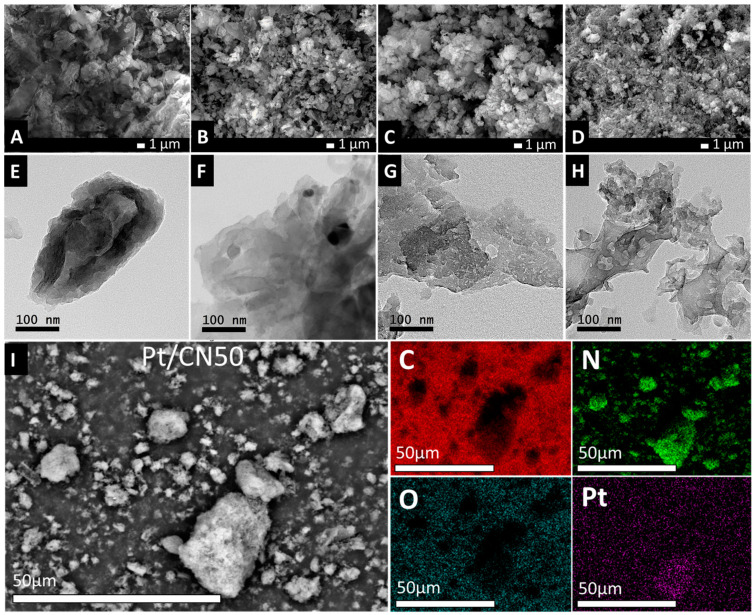
SEM images of Pt/CN (**A**), Pt/CN0 (**B**), Pt/CN50 (**C**), Pt/CN100 (**D**) and TEM images of Pt/CN (**E**), Pt/CN0 (**F**), Pt/CN50 (**G**), Pt/CN100 (**H**); and SEM image of Pt/CN50 (**I**) with EDS mapping of elements C, N, O and Pt.

**Figure 2 nanomaterials-12-01188-f002:**
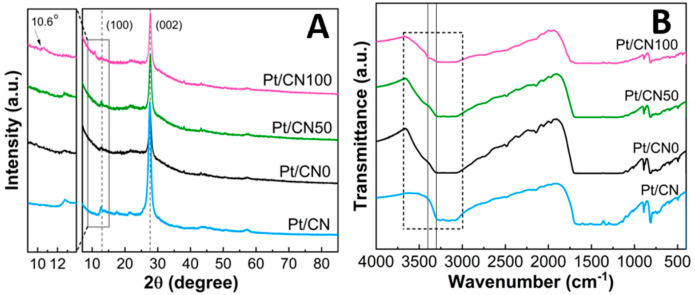
(**A**) XRD patterns and (**B**) FT–IR spectra of Pt/CNx photocatalysts.

**Figure 3 nanomaterials-12-01188-f003:**
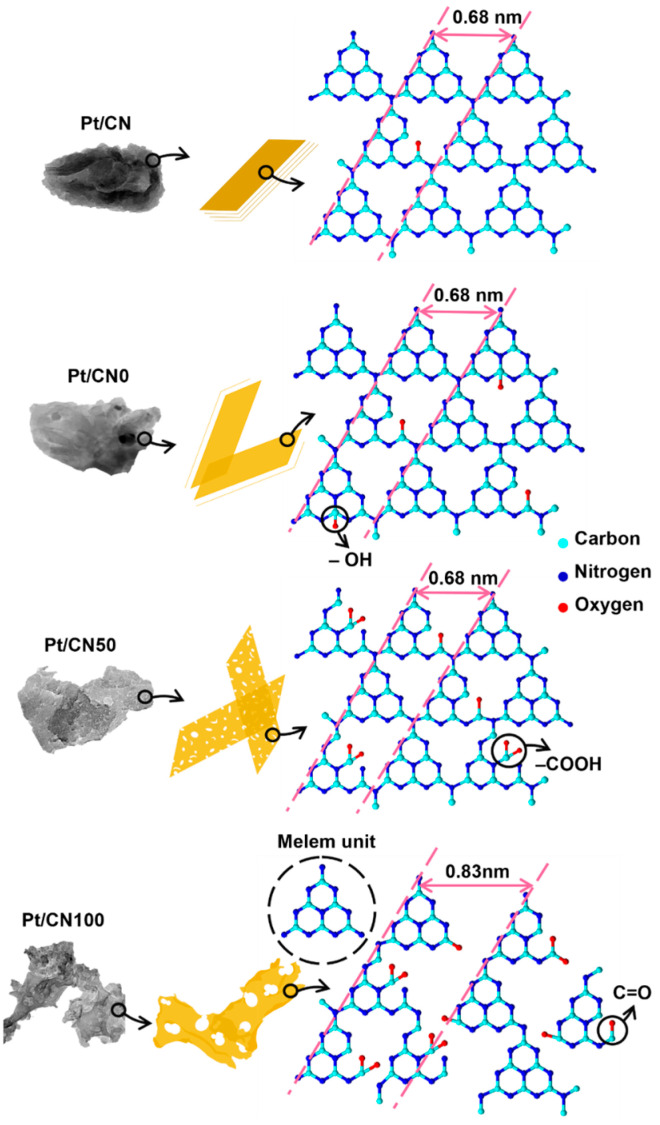
A schematic illustration of change in the structural and physicochemical properties of g-C_3_N_4_ surface by the solvent etching process.

**Figure 4 nanomaterials-12-01188-f004:**
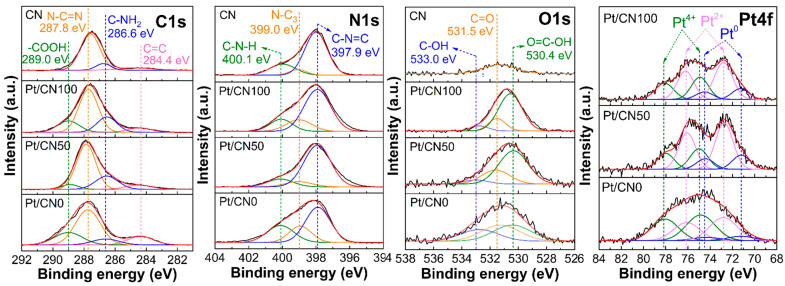
XPS data of C 1s, N 1s, O 1s, and Pt 4f for all the photocatalysts.

**Figure 5 nanomaterials-12-01188-f005:**
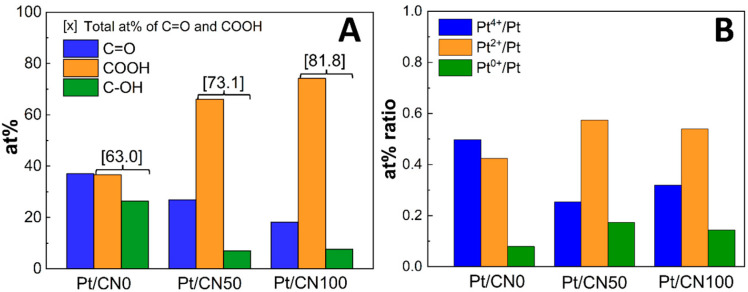
(**A**) the at% of oxygen functional groups and (**B**) at% of platinum at various oxidation stage in prepared samples calculated from XPS data.

**Figure 6 nanomaterials-12-01188-f006:**
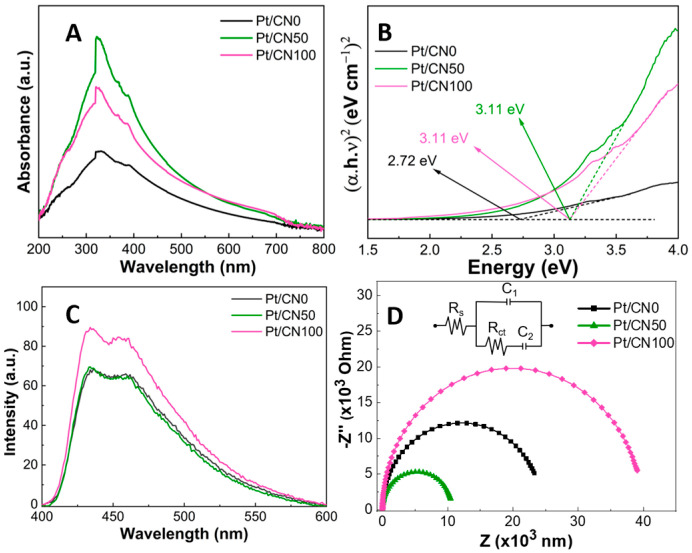
(**A**) UV-vis absorption spectra, (**B**) Tauc plots, (**C**) PL spectra, and (**D**) The EIS Nyquist plots of the Pt/CNx photocatalysts.

**Figure 7 nanomaterials-12-01188-f007:**
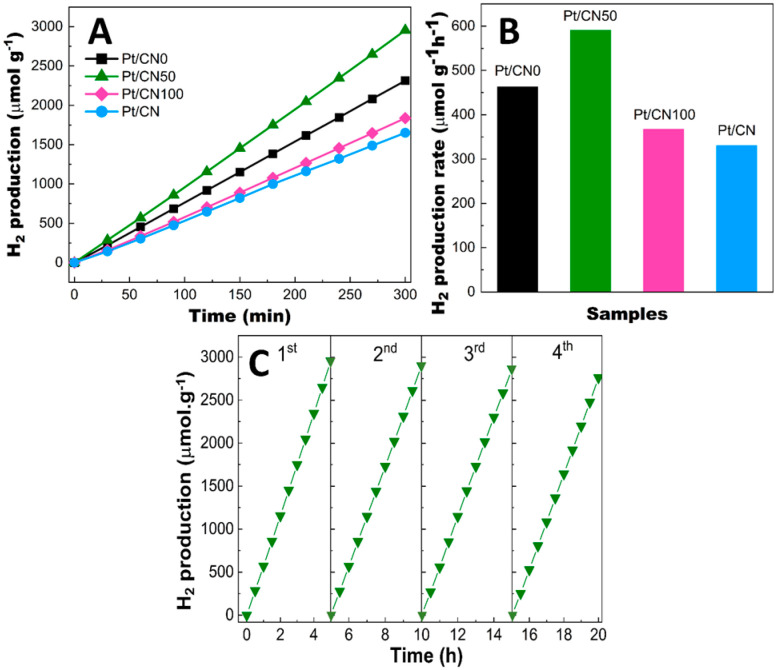
(**A**) Photocatalytic hydrogen evolution over time, (**B**) H_2_ production rate of all the photocatalysts, and (**C**) H_2_ production recycling of Pt/CN50.

**Figure 8 nanomaterials-12-01188-f008:**
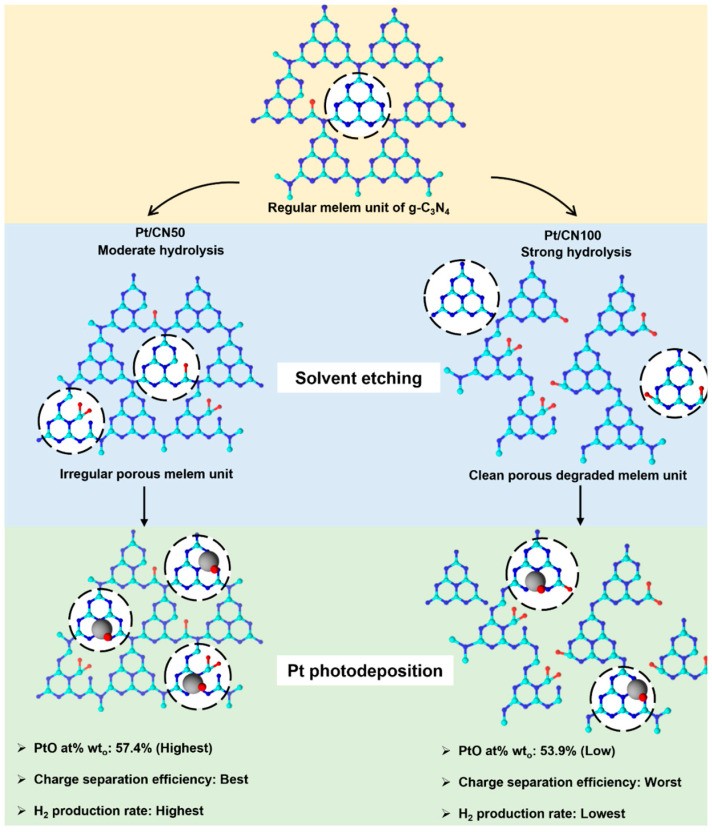
A schematic diagram for key properties of Pt/CN50 and Pt/CN100 in each preparation step.

**Table 1 nanomaterials-12-01188-t001:** Pt (%) content, textural data and band gaps of the prepared photocatalysts.

Sample	Pt (%) ^a^	S_BET_ (m^2^/g) ^b^	V (cm^3^/g) ^b^	L (nm) ^b^	Band Gap (eV) ^c^
Pt/CN	3.24	36.3	0.254	27.8	2.58
Pt/CN0	2.29	56.3	0.483	3.83	2.72
Pt/CN50	1.54	70.5	0.435	3.82	3.11
Pt/CN100	3.03	51.6	0.424	24	3.11

^a^ Obtained from ICP measurement. ^b^ Specific surface area, pore volume, and average pore size were determined via N_2_ adsorption–desorption isotherm measurements. ^c^ Estimated band gaps were obtained from the Tauc plots of the UV-Vis spectra.

**Table 2 nanomaterials-12-01188-t002:** O/N atomic ratios of all prepared samples obtained from elemental analysis.

Element Analysis				
Samples	Atom (wt%)			Atomic Ratio
	C	N	O	O/N
Pt/CN	35.7	59.1	0.46	0.008
Pt/CN0	33.9	63.6	0.61	0.008
Pt/CN50	33.5	62.2	2.05	0.029
Pt/CN100	31.6	60.6	4.71	0.068

## Data Availability

The data presented in this study are available on request from the corresponding author.

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
