# Peer review of "Solvent Etching Process for Graphitic Carbon Nitride Photocatalysts Containing Platinum Cocatalyst: Effects of Water Hydrolysis on Photocatalytic Properties and Hydrogen Evolution Behaviors"

_nanomaterials, 2022, doi:10.3390/nano12071188_

Round 1
Reviewer 1 Report
This manuscript reports a facile solvent etching process on g-C3N4 via 50:50 ethanol/water solvent to optimize its surface properties and the visible light photocatalytic hydrogen evolution activity of modified g-C3N4. In recent years, the etching modification of carbon nitride materials has been widely studied, and the hydrothermal solvent etching method is the most common one. Although there is a lack of the novelty of the topic in this manuscript, the perspective of analysis and explanation has certain reference values. The moderate hydrolysis effect of the solvent mixture generated ample O-containing functional groups on the surface of pristine g-C3N4, resulting in the efficient charge transfer and the slow recombination of electron-hole pairs. This phenomenon positively influenced the oxidation state of the Pt cocatalyst in the photodeposition process and promoted the rate of hydrogen production. Considering these points and the detailed discussion, I recommend that this manuscript can be published after the following revisions.
- Why the amount of Pt cocatalyst loaded on Pt/CN50 sample is much lower than the others according to Table 1? Is this difference considered in the final conclusions? Especially the effect on the Pt particle sizes and hydrogen evolution performance due to agglomeration of metal particles.
- The authors characterized the photogenerated electron-hole pair separation efficiency via PL emission spectra. Please explain clearly the attribution of the two peaks (435 and 460 nm) among three samples in Figure 6c.
- It's commonly considered that the polymer carbon nitride is a kind of direct-gap semiconductor according to the band diagram. So, the Y-axis title in Figure 6b needs to be corrected.
- The reason for the better activity of hydrogen production over Pt/CN50 cannot be simply attributed to the behavior of the photogenerated carriers and the oxidation state of Pt. It would be better if the authors explain the effect of specific surface areas and the size of Pt particles.
- It is also better to cite the following papers as references:
1) Angew. Chem. Int. Ed., 2022, 61, e202113389. DOI: 10.1002/anie.202113389
2) Chem. Commun., 2021, 57, 4138. DOI: 10.1039/d1cc00214g
3) Angew. Chem. Int. Ed., 2019, 58, 14950–14954. DOI: 10.1002/ange.201908322
4) Appl. Catal. B., 298 (2021) 120565. DOI: 10.1016/j.apcatb.2021.120565
Reviewer 2 Report
- The abstract section is poorly constructed. No clear information about the examined catalytic materials
- The authors should give more precise details about the purpose of the work in the introduction section. The novelty or significance of this work is not clearly stated.
- Many space errors/punctuation errors must be solved. The abbreviations should be checked in the manuscript and make clear.
- XRD patterns must be labeled.
- Figure X needs to be discussed be uniformly in the manuscript.
- Nyquist plot must be described in detail with equivalent circuit and fitted parameters.
- After stability test- Morphological and surface examination of material must be described.
- During the presentation of the results, compare HER performance with the latest literature. After presenting the results, it is recommended to provide a short conclusion to the obtained results.
- There are many errors in the paper, so the Authors are encouraged to review the form and the manuscript's English.
Reviewer 3 Report
The manuscript entitled ‘Solvent Etching Process For Graphitic Carbon Nitride Photo-catalysts Containing Platinum Cocatalyst: Effects Of Water Hydrolysis On Photocatalytic Properties And Hydrogen Evolution Behavior' by Hoang et al. describes the surface etching of carbon nitride photocatalysts to enhance the surface area and introduce surface functional groups in an attempt to increase the photocatalytic hydrogen production. The work is interesting in the context of organic based photocatalyts. However, the following issues should be checked and addressed prior to publication to insure the quality of the work:
1.The authors use Pt as hydrogen evolution reaction catalyst, and it was deposited on the CN using photodepostion. However, the use of only DI water in the deposition will lead to the hydrolysis of the Pt precursor to form the respective Pt oxides not the metallic Pt nanoparticles which are catalytically active. Conventionally the photodeposition is performed in a solvent environment with hole acceptors/ scavengers to obtain the metallic state. Thus the authors need to address what benefits such photodeposition approach has on the established methods.
2. My comments in 1 is supported by the XPS data presented by the authors. It confirms the predominant formation of Pt oxides instead of metallic Pt. Curiously, the O1s XPS signal doesn’t show the lattice oxygen from Pt/ or the authors overlooked the data and the assignment is missing. Generally, Pt oxides are expected to exhibit less catalytic activity for hydrogen evolution than the Pt metal (which is true also in this study Fig. 7a with some anomalies). The authors need to check the XPS data carefully and also present the role of ‘ Pt oxides’ in relation to the current system.
3. EIS measurements can give estimation of film conductivity/ resistivity but to explain the results in terms of charge transfer dynamics / charge carrier recombination it should be performed in the presence of redox couple or under light irradiation. Care should be taken in the interpretation of the results.
4. Still it is not clear the observed trend in the photocatalytic properties originate from the structural and morphological difference of the CN or the different ‘Pt oxide ratios’. Since most of the physico chemical characterizations were performed on the Pt-modified CN catalysts, control experiments on the pristine CN will give more insights to the study.
Round 2
Reviewer 3 Report
I acknowledge the effort of the authors to address some of the issues, however, there are still concerns and should be addressed carefully,
- Generally, the authors respond to reviewers’ comments in detail but changes in the manuscript are minor.
- The Pt4f XPS signal show Pt both +4 and +2 oxidation states and suggesting two Pt oxide species, so this should be indicated on the O1s signal and included in the manuscript. If the authors reported similar systems in their previous publication (which clearly show the Pt oxides), why the O1s show a different signature in the current study?
- The measurement condition for the EIS should be included, i.e. if it is performed under illumination it should be stated in the experimental. Additionally, the fitting should be checked. The values extracted from the fitting do not correlate the Nyquist diagram. According to Fig. 6D the lowest charge transfer resistance is expected for the Pt/CN50 not for Pt/CN100 (see the values table R1).
- Still the control experiments on the pristine CN are recommended.
